# Pathogenesis of Paradoxical Reactions Associated with Targeted Biologic Agents for Inflammatory Skin Diseases

**DOI:** 10.3390/biomedicines10071485

**Published:** 2022-06-23

**Authors:** Fumi Miyagawa

**Affiliations:** Department of Dermatology, Nara Medical University School of Medicine, Nara 634-8522, Japan; fumim@naramed-u.ac.jp; Tel.: +81-744-29-8891; Fax: +81-744-25-8511

**Keywords:** biologic agent, paradoxical reaction, psoriasis, atopic dermatitis, cytokine

## Abstract

Targeted biologic agents have dramatically changed the therapeutic landscape for immune-mediated inflammatory diseases, particularly in rheumatology and dermatology. Their introduction has resulted in a paradigm shift, i.e., they produce significant clinical improvements in most patients with such diseases. Nevertheless, a variety of adverse reactions associated with these agents have been observed, including so-called paradoxical reactions (PRs), which are a new class of adverse events. PRs involve the de novo development or worsening of immune-mediated inflammatory disease during treatment with a targeted biologic agent that is commonly used to treat the idiopathic counterpart of the drug-induced reaction. In addition, the efficacy of biologic agents targeting individual cytokines and the existence of PRs to them have provided proof that cytokines are key drivers of various immune-mediated inflammatory diseases and helped researchers elucidate the molecular pathways underlying the pathophysiology of these diseases. Here, a comprehensive review of the targeted biologic agents used to treat immune-mediated inflammatory diseases, particularly psoriasis and atopic dermatitis, is provided, with a specific focus on biologic agents that inhibit cytokine signaling involving tumor necrosis factor-α, interleukin (IL)-12/23 (p40), IL-17A (and the IL-17 receptor [R]), IL-23 (p19), and the IL-4Rα, and their associated PRs. The characteristic clinical manifestations and potential immunological mechanisms of the PRs induced by these biologic agents are also reviewed.

## 1. Introduction

With recent advances in our molecular understanding of the immune axes activated in various immune-mediated inflammatory diseases, targeting pathogenic cytokines with greater specificity has become possible. Biologic agents, such as tumor necrosis factor (TNF)-α inhibitors, are used as therapies that target a specific molecule of the immune system. These treatments have revolutionized medicine, offering targeted therapy for an increasing number of immune-mediated inflammatory diseases, including rheumatoid arthritis (RA), inflammatory bowel disease, and psoriasis. Although they have significantly improved outcomes and are widely used, unexpected adverse effects—so-called paradoxical reactions (PRs)—arise in some patients [1,2,3]. PRs encompass a variety of immune-mediated inflammatory reactions, which manifest unexpectedly after the administration of a biologic agent. These include the de novo development or worsening of an immune-mediated disease and the worsening of the condition being treated with the biologic agent [1,2]. The skin is frequently affected, with a wide spectrum of clinical and histological reaction patterns seen. Among them, psoriatic eruptions induced by TNF-α inhibitors is the prototypic and most frequently observed PR to biologic agents [2].

This article reviews the recent literature on PRs associated with TNF-α, interleukin (IL)-12/23 (p40), IL-17A/the IL-17 receptor (R), IL-23 (p19), and IL-4Rα inhibitors, which are commonly used for dermatological conditions, especially atopic dermatitis (AD) and psoriasis. Psoriasiform and eczematous eruptions are the most common PRs, but many other rare phenotypes have been described, including lupus-like reactions, alopecia areata, lichen planus, hidradenitis suppurativa, pyoderma gangrenosum, sarcoidosis-like reactions, vitiligo, and bullous pemphigoid [1,2]. The most commonly proposed mechanism for explaining the PRs of biologic agents is that the outcome of the immune response is strictly dependent on the balance between cytokine patterns; therefore, inhibiting subsets of cytokines using biologic agents can induce an imbalance favoring another immune response pattern [4]. Here, we summarize current proposals regarding the detailed pathomechanisms of the psoriasiform and eczematous eruptions and lupus-like reactions induced by the five abovementioned classes of biological agents.

## 2. Signature Cytokines and Associated Molecular Pathways Involved in the Idiopathic Counterparts of PRs Caused by Biologic Agents

Advances in the pathogenic characterization of immune-mediated inflammatory diseases have resulted in the development of molecular-based classifications [4,5,6]. The most well-known classification is that immune responses may be categorized into three major types (type 1, type 2, and type 3 responses) on the basis of the specific subsets of innate lymphoid cells (ILCs), innate effector cells, and CD4 effector T cells involved [7]. In each type of immune response, the innate and adaptive immune systems act together to eliminate a specific type of pathogen. Type 1 immunity provides protection against intracellular microbes, such as bacteria, protozoa, and some viruses. Type 2 immunity is mainly devoted to protection against helminths and venoms, and type 3 immunity targets extracellular bacteria and fungi (Figure 1). However, inappropriate activation of a specific type of immune response can cause different types of immune-mediated disorders (Figure 1). For example, interferon (IFN)-associated type 1 immune responses may be directed against self-antigens, leading to autoimmune diseases, including lupus erythematosus and lichen planus [6,8]. On the other hand, exaggerated type 2 immunity is associated with allergic diseases, such as asthma and AD, and dominant type 3 immunity is detected in psoriasis (Figure 1). Interestingly, the key cytokines of one immune pathway are capable of blocking another pathway. For instance, AD and psoriasis represent two distinct pathogenetic entities at polar ends of the T-cell inflammatory milieu; AD is considered to be a polar T helper 2 (Th2) cell disease, while psoriasis is considered to be driven by Th1/Th17 cells, and they usually mutually antagonize each other [9]. Grouping immune-mediated inflammatory diseases according to their molecular pathogenesis also provides a rationale for the use of targeted therapies [4,5,6]. In this section, the pathogeneses of the idiopathic counterparts of PRs caused by biologic agents (psoriasis, AD, and lupus) are described, with specific focus on the signature cytokines involved in each disease.

### 2.1. IL-17 and IL-23 in Psoriasis

Psoriasis is a T-cell-mediated disease, which is primarily driven by pathogenic Th17 cells that produce high levels of IL-17 in response to IL-23 [10]. Lowes et al. detected higher IL-17A mRNA expression in psoriatic skin than in non-lesioned skin [11]. Furthermore, T cells obtained from the dermis and peripheral blood in psoriasis patients produce IL-17 [11]. In addition, a mouse strain that expressed transgenic IL-17A in its skin developed severe psoriasis-like skin inflammation, suggesting that this cytokine plays an important role in psoriasis [12].

The expression of IL-23 was also examined in the skin lesions of psoriasis patients, and large increases in the copy numbers of mRNAs encoding IL-23p19 and p40, but not IL-12p35, were detected [13]. Immunostaining showed significantly higher IL-23p19 expression by keratinocytes in psoriatic skin lesions than by keratinocytes from normal or non-lesional psoriatic skin [14]. Double immunostaining also demonstrated that IL-23p19 is expressed by epidermal Langerhans cells, dermal dendritic cells (DCs), and macrophages [14]. In addition, transgenic mice that expressed IL-23 in their skin developed psoriasis and psoriatic arthritis [15]. In addition to these findings, the superiority of IL-17/IL-23 inhibitors over traditional systemic agents underscores the central role played by the IL-23/Th17 cell axis in psoriasis [10]. Lowes et al. also demonstrated that IFN-γ mRNA expression was increased in psoriatic skin, and there were discrete populations of T cells producing IFN-γ and TNF-α in the dermis and circulation of psoriasis patients, suggesting that psoriasis involves a mixed Th1 and Th17 inflammatory environment [11]. The upregulation of TNF-α expression in psoriatic skin has also been well characterized, and it was found that TNF-α was expressed almost exclusively in TNF- and inducible nitric oxide synthase (iNOS)-producing DCs [16].

Using a xenograft model of human psoriasis, in which psoriatic skin lesions spontaneously developed on the pre-psoriatic skin of psoriasis patients after it was transplanted into AGR129 mice, Boyman et al. demonstrated that TNF-α plays an essential role in local T-cell proliferation and the development of psoriatic plaques [17]. In this model, immunostaining showed that antigen-presenting cells were the predominant source of TNF-α, and neutralizing TNF-α resulted in decreased psoriasis development and a significant reduction in the number of T cells in the graft [17]. Subsequently, Nestle et al. demonstrated the contribution of IFN-α produced by dermal plasmacytoid DCs (pDCs) to the initiation of psoriatic lesions [18]. They showed that pDCs infiltrate the skin of psoriatic patients and become activated to produce IFN-α early during disease formation. Using the xenograft model of human psoriasis mentioned above, they demonstrated that blocking IFN-α signaling by administering neutralizing anti-IFN-α/β receptor antibodies or inhibiting the ability of pDCs to produce IFN-α by injecting anti-BDCA-2 monoclonal antibodies (mAbs) prevented the development of psoriasis [18]. Furthermore, the addition of IFN-α induced the expansion of pathogenic T cells and the development of psoriasis in vivo. Collectively, these results indicate that IFN-α production by dermal pDCs is a key element in the early phase of psoriatic skin lesion induction [18] (Figure 2).

It has been hypothesized that the IFN-α and TNF-α produced by pDCs in psoriatic skin drives IL-23 production by myeloid DCs and the subsequent activation of Th17 cells. TNF-α also acts synergistically with IL-17 to upregulate the expression of inflammatory genes implicated in psoriasis, including genes that encode key molecules in the molecular signature of psoriasis, such as psoriasin (S100A7), β-defensin, IL-8, chemokine (C-C motif) ligand 20 (CCL20), IL-23 (p19), and chemokine (C-X-C motif) ligand 1 (CXCL1) [19]. The efficacy of TNF-α inhibitors against psoriasis underscores the importance of this cytokine in the disease, although the percentage of patients that experience marked improvements in their conditions is significantly lower than is the case after treatment with IL-17 or IL-23 inhibitors [10].

### 2.2. IL-4, IL-13, and IL-5 in Atopic Dermatitis

Th2 cytokines or type 2 cytokines, including IL-4, IL-13, and IL-5, constitute the signature cytokines that drive the pathogenesis of AD through their effects on a variety of immune and epithelial cells (Figure 1). It is well known that immune responses in AD lesions deviate toward type 2 immune responses, as AD lesions contain a significantly increased number of Th2 cytokine mRNA-expressing cells [20], and transgenic mice expressing Th2 cytokines developed AD lesions [21,22,23]. Investigations of transcriptomes also revealed that AD lesions are associated with Th2 immune axes [24]. The fact that dupilumab substantially improves the symptoms of moderate-to-severe AD supports the idea that IL-4 and IL-13 play crucial roles in the pathogenesis of AD.

### 2.3. Type I IFNs in Systemic Lupus Erythematosus

It is well established that type I IFNs (IFN-α/β) play a major pathogenic role in systemic lupus erythematosus (SLE) [25]. Patients with active SLE display increased serum levels of type I IFNs, which correlate well with disease activity [26,27] and the subsequent increases in the expression levels of IFN-stimulated genes (ISGs) (the IFN signature) seen in their peripheral blood mononuclear cells (PBMCs) [28,29]. Further confirmation of the role played by type I IFNs in the pathogenesis of SLE came from case reports describing the onset of SLE during IFN-α therapy [30,31]. Both in vivo [32] and in vitro [33] studies have shown that the hyperproduction of type I IFNs by pDCs makes a critical contribution to the pathogenesis of SLE. Studies in lupus-prone mice also demonstrated the importance of type I IFNs for SLE because NZB mice lacking the α-chain of the type I IFN receptor exhibited significantly decreased morbidity and mortality [34]. Furthermore, mice that were deficient in IFN regulatory factor 7, a master regulator of type I IFN-dependent immune responses, failed to produce autoantibodies when SLE was chemically induced [35,36]. Collectively, these findings underscore the role played by type I IFNs in the pathogenesis of SLE.

## 3. Paradoxical Reactions Caused by Each Type of Biologic Agent

### 3.1. TNF-α Inhibitors

TNF-α is a pleiotropic cytokine which plays major roles in inflammatory responses and immune regulation. It is now accepted that TNF-α plays a critical role in the pathogeneses of certain autoimmune diseases, including RA and psoriasis. TNF-α inhibitors were introduced more than 20 years ago and are being used for an expanding number of rheumatic and autoimmune diseases, such as RA, inflammatory bowel disease (ulcerative colitis and Crohn’s disease), psoriasis, and ankylosing spondylitis. Five TNF-α inhibitors are currently available: infliximab, adalimumab, etanercept, certolizumab pegol, and golimumab (Table 1). Since TNF-α inhibitors were the first biologics to be developed, adverse cutaneous reactions to their use have been extensively investigated, and numerous inflammatory and autoimmune-like cutaneous reactions have been reported during anti-TNF-α therapy [37]. Psoriasiform and eczematous reactions are the most common cutaneous adverse reactions associated with TNF-α inhibitors. Other inflammatory cutaneous side effects include lupus-like disorders, vasculitis, sarcoidosis-like and other granulomatous reactions, and lichenoid reactions, but these appear at lower frequencies [1,38].

The primary mechanism of action of TNF-α inhibitors in patients with psoriasis most likely involves the indirect inhibition of IL-17 signaling through the regulation of IL-23 production by myeloid DCs [39]. Using microarray analysis, Zaba et al. demonstrated that the efficacy of TNF-α inhibitors is linked to the suppression of IL-17A, as the rapid downregulation of IL-17A pathway genes was seen in patients who responded to etanercept, but not in non-responders [40].

#### 3.1.1. Psoriasiform Reactions

Despite the efficacy of TNF-α inhibitors against psoriasis, the onset or worsening of psoriasis has been reported in patients treated with TNF-α inhibitors for various systemic inflammatory rheumatic diseases in numerous studies. Psoriasiform eruptions are the most widely reported cutaneous PRs associated with TNF-α inhibitors. An international registry of autoimmune diseases induced by biologics (the BIOGEAS registry), which included 12,731 patients, demonstrated that psoriasis was the most frequently induced autoimmune disorder (6375 cases), and 99% of cases were caused by TNF-α inhibitors [3].

The association between TNF-α inhibitors and psoriasiform eruptions was first suggested in a study of a cohort of 107 patients with spondyloarthropathy that were treated with infliximab, in which three patients without a personal or family history of psoriasis developed palmoplantar pustulosis [41]. Around the same time, several cases of psoriasiform eruptions induced by TNF-α inhibitors were reported in small case series [42,43,44,45]. Furthermore, a systematic review of 207 published cases of TNF-α inhibitor-induced psoriasiform eruptions demonstrated that the risk of such adverse events is not affected by the underlying disease treated with the TNF-α inhibitor, which included psoriasis, psoriatic arthritis, RA, inflammatory bowel disease, ankylosing spondylitis, Behcet’s disease, and juvenile idiopathic arthritis [46]. Of these cases, 43%, 26%, and 20% involved RA, seronegative spondyloarthropathy, and inflammatory bowel disease, respectively. The risk of TNF-α-induced psoriasiform eruptions also appeared to be independent of the type of TNF-α inhibitor used. Fifty-nine percent, 22%, and 19% of patients were being treated with infliximab, adalimumab, and etanercept, respectively. Similarly, another systematic review, including 36 studies, demonstrated that infliximab was the most common causative drug (56.6% of cases), with adalimumab and etanercept accounting for 30% and 11% of cases, respectively, and certolizumab and golimumab accounting for <2.5% of cases [1]. Morphologically, the patients were described as having pustular psoriasis in 56% of cases, plaque psoriasis in 50% of cases, and guttate lesions in 12% of cases; 15% of cases involved more than one type of lesion [46]. The clinical characteristics of TNF-α inhibitor-induced psoriasiform eruptions include a higher frequency of palmoplantar involvement compared with classical psoriasis [47].

The incidence of psoriasiform eruptions caused by TNF-α inhibitors was examined in a large cohort of 9826 RA patients treated with TNF-α inhibitors from the British Society for Biologics Registrar [48]. This study described 25 incident cases of psoriasis, and an increased incidence of psoriasiform reactions was observed among the patients treated with TNF-α inhibitors (1.04 per 1000 person years) compared with that seen in the patients treated with traditional disease-modifying antirheumatic drugs (0 per person per year) [48]. Furthermore, the patients treated with adalimumab demonstrated a higher rate of incident psoriasis than those treated with etanercept or infliximab [48]. The estimated incidence of psoriasiform eruptions caused by TNF-α inhibitors ranges from 3.5% to 10.7% [49,50]. The time to onset varies, ranging from <1 month to >10 years, with an average of 16.4 years [1].

##### Mechanism Responsible for Psoriasiform Reactions Induced by TNF-α Inhibitors

Given that TNF-α is a key regulator of psoriasis development, the idea of TNF-α inhibitor-induced psoriasis seems to contradict the therapeutic benefits of TNF-α inhibitors against psoriasis, and the underlying pathophysiological mechanisms responsible for TNF-α inhibitor-induced psoriasis remain elusive. The cross-regulation of TNF-α and IFN-α was suggested to play a role by Palucka et al., who demonstrated that TNF-α regulates IFN-α production (Figure 2), and blocking endogenous TNF-α maintained IFN-α production by pDCs [51]. Thus, TNF-α blockade by TNF-α inhibitors may induce sustained IFN-α production, leading to the development of psoriasis. Consistent with this hypothesis, TNF-α inhibition has been shown to induce increased expression of IFN-stimulated genes in patients with systemic-onset juvenile idiopathic arthritis [51], and immunohistochemical staining of skin biopsy specimens for myxovirus-resistance protein A (MxA), a surrogate marker of lesional type I IFN activity, showed greater staining in TNF-α inhibitor-induced psoriasis than in psoriasis vulgaris [52]. Furthermore, by analyzing mRNA expression, Conrad et al. demonstrated that the expression of the type I IFNs *IFNA2* and *IFNB1* was significantly greater in skin lesions caused by TNF-α inhibitor-induced psoriasis than in those caused by chronic plaque psoriasis, while no significant difference in the expression levels of other cytokines, such as *TNF*, *IL23A*, *IL17A*, *IL17F*, and *IL17C*, were detected between skin lesions caused by TNF-α inhibitor-induced psoriasis and those caused by classical psoriasis [47]. They also demonstrated that the skin lesions caused by TNF-α inhibitor-induced psoriasis are characterized by the dermal accumulation of pDCs and that TNF-α blockade prolongs the ability of pDCs to produce IFN-α by inhibiting their maturation, which results in type I IFN overexpression [47]. However, anti-IFN-α mAbs failed to ameliorate psoriasis in a clinical trial, challenging the idea that IFN-α plays a central role in psoriasis, at least in established chronic plaque psoriasis [53].

#### 3.1.2. Eczematous Reactions

Eczematous eruptions have been reported after the use of TNF-α inhibitors for various rheumatic diseases. The first cases were reported by Wright et al., in which two RA patients treated with infliximab developed AD-like eruptions [54]. The first large prospective study was conducted by Flendrie et al., which included 289 patients with RA who were treated with TNF-α inhibitors, focused on cutaneous reactions [55]. Of these, 20 cases of eczematous eruptions were documented, 5 of which were histopathologically confirmed by a skin biopsy [55]. Another prospective study, which included 150 patients with rheumatic diseases that were treated with TNF-α inhibitors, demonstrated that 8 patients developed eczematous eruptions [56]. A literature search revealed that eczematous eruptions have been reported in 5–20% of patients that were treated with TNF-α inhibitors for various inflammatory diseases, with infliximab being the inhibitor that was most strongly associated with such adverse reactions [57].

##### Mechanism Responsible for Eczematous Reactions Induced by TNF-α Inhibitors

The precise mechanism by which TNF-α inhibitors cause eczematous eruptions remains unclear. Their mechanism of action is considered to involve alterations in the cytokine balance and the suppression of Th17/Th1 cells. It has been demonstrated that the treatment of psoriasis patients with etanercept rapidly downmodulated the levels of Th17 cell products and subsequently reduced the levels of Th1 cell products [58]. In addition, IL-4 expression was upregulated months after the disease had been significantly ameliorated, indicating that etanercept also modulates Th2 cell expression [58]. Similarly, Quaglino et al. investigated the changes in the frequencies of CD4 T-cell subsets induced by etanercept in psoriasis patients [59]. The upregulation of Th1/Th17 subsets and downregulation of regulatory T-cell (Treg) subsets was observed before treatment. The patients who responded to etanercept showed a significant downregulation of Th1/Th17 subsets and a concomitant upregulation of Th2 and Treg subsets [59]. Indeed, treatment with TNF-α inhibitors has been reported to induce eosinophilia in psoriasis patients, suggesting that they revert to a Th2 phenotype [60]. These findings suggest that TNF antagonism may induce an immune deviation from the Th1/Th17 phenotype to the Th2 phenotype and that CD4 T-cell modulation can be effective in reversing the psoriatic phenotype, as IL-4 therapy produced a marked clinical improvement by increasing the number of IL-4-producing CD4 T cells in psoriasis patients [61].

However, a detailed characterization of the immune phenotype of TNF-α inhibitor-induced psoriasiform and eczematous eruptions revealed that these lesions were immunologically different from those found in conventional psoriasis and eczema [62]. By performing histopathological evaluations and gene expression and computer-assisted immunohistological studies on skin biopsy samples from TNF-α inhibitor-induced skin lesions, Stoffel et al. found increased IFN-α and Th1 cytokine expression, particularly IFN-γ expression, in both eczematous and psoriasiform eruptions caused by TNF-α inhibitors, whereas Th17 predominated in conventional psoriasis, and Th2 predominated in conventional atopic eczema [62]. In addition, the in vitro culturing of PBMCs with different IFN-α subtypes revealed that IFN-α-5 significantly increased the percentage of Th2 cells, suggesting that a subtype of IFN-α is linked to Th2 polarization [63].

In another study, a personal history of atopy was found to be the only predictor of the occurrence of eczematous eruptions in patients receiving infliximab (odds ratio = 3.6), and other factors, including sex, age, the principal diagnosis, the dose and duration of infliximab therapy, and the concomitant use of immunosuppressive therapies, had no influence on their occurrence [64].

#### 3.1.3. Lupus-Like Reactions

A well-described side effect of TNF-α inhibitors is the development of humoral autoimmunity in some patients, which is characterized by the production of antinuclear antibodies (ANAs) and anti-dsDNA antibodies. Several lines of evidence have indicated that a significant proportion of patients treated with TNF-α inhibitors develop these antibodies, regardless of their underlying disease. Less commonly, patients treated with TNF-α inhibitors develop clinical features of SLE. The clinical presentations of such cases vary from SLE, to lupus-like syndromes, to isolated cutaneous lupus [1]. In a prospective single-center cohort study of 454 RA patients treated with TNF-α inhibitors, ANAs developed in 31.2% of infliximab-treated patients, 16.1% of those treated with adalimumab, and 11.8% of those treated with etanercept [65]. The median treatment duration prior to ANA seroconversion was 10.9 (1.3–80.0) months. Of the 83 patients that seroconverted to ANA positivity, 6 (7.2%) became positive for anti-dsDNA antibodies. In the latter study, only 3 patients developed drug-induced lupus (DIL) although one of them was already positive for ANAs before treatment [65]. Another study demonstrated that ANA positivity was observed in 63.1%, 51.1%, and 44.8% of RA patients at 24 weeks during treatment with infliximab, etanercept, and adalimumab, respectively [66].

Despite the relatively high rate of ANA positivity, only a small proportion of patients develop DIL, i.e., the frequency of DIL was <1% in most studies, and infliximab, which is a chimeric-murine mAb, is the TNF-α inhibitor that most commonly causes DIL. For instance, an international registry of autoimmune diseases induced by biologics (the BIOGEAS registry) demonstrated that 0.33% of the patients exposed to biologics (97% of them were TNF-α inhibitors) developed DIL, and the frequency of DIL was higher in patients with RA (0.5%) and those treated with infliximab (0.66% vs. 0.49% for etanercept and 0.11% for adalimumab) [3]. Another study of a nationwide pharmacovigilance database showed that among 5213 cases of adverse drug reactions to TNF-α inhibitors, DIL accounted for 39 (0.75%): 25 involved infliximab, 9 involved adalimumab, and 5 involved etanercept [67]. A systematic review of published cases demonstrated that the most commonly reported causative drugs of lupus-like reactions are infliximab (56%), adalimumab (25%), and etanercept (15.5%), with incidence rates of 0.175%, 0.06%, and 0.09%, respectively [1]. It also demonstrated that the mean time to the onset of lupus-like reactions was 14.6 months but ranged from <1 month to >6 years [1]. Other studies reported that the median time to the onset of lupus was 11 months [67] or 14 months [68]. However, favorable outcomes are usually achieved. The vast majority of patients achieve complete or partial resolution of their lupus-like reactions upon the withdrawal of TNF-α inhibitors, with a subset requiring ongoing lupus-specific therapy [67,68].

##### Clinical Features of Lupus-like Reactions Induced by TNF-α Inhibitors

DIL that occurs secondary to TNF-α inhibitor treatment frequently produces a fever, arthritis, arthralgia, and/or a skin rash, and it only rarely causes hematological, renal, or central nervous system manifestations. Patients with such DIL are invariably positive for ANAs, and positivity for anti-dsDNA is relatively common. In addition, hypocomplementemia is not rare [69,70]. Only 16.7–35% of patients fulfil the criteria for SLE [3,67,68,70]. The skin is affected in 89% of cases caused by TNF-α inhibitors, with the cutaneous manifestations including a malar rash, photosensitivity, and discoid and subacute cutaneous lupus [67,68,70]. Among such cases, cutaneous involvement was the sole manifestation in 56% of patients [68].

##### Mechanism Responsible for Lupus-like Reactions Induced by TNF-α Inhibitors

The mechanism responsible for the development of lupus-like reactions in patients treated with TNF-α inhibitors has not been established. Since SLE is recognized to be driven by type I IFNs (IFN-α/β), Palucka et al. investigated whether reciprocal regulation of TNF-α and IFN-α is involved in such reactions [51]. They reported that the patients treated with TNF-α inhibitors displayed increased transcription of IFN-stimulated genes. Furthermore, in vitro studies showed that TNF-α inhibits both the generation of pDCs from hematopoietic progenitors, and virus-induced IFN-α release from immature pDCs by inducing their maturation [51]. They also demonstrated that the neutralization of endogenous TNF-α helps to maintain IFN-α secretion by pDCs [51]. These results suggest that TNF-α downregulates IFN-α expression under normal circumstances, and TNF-α may act as an antagonist of the type I IFN pathway. Through this mechanism, TNF-α inhibitors can cause an unregulated increase in IFN-α and, hence, induce lupus-like reactions.

Using lupus-prone model mice, the (NZBxNZW)F1 system, the physiological functions of TNF-α have been investigated [71]. In the latter study, restriction fragment length polymorphism analysis revealed reduced levels of TNF-α production in the NZW mice, and TNF-α replacement therapy induced a significant delay in the development of nephritis in (NZBxNZW) F1 mice [71]. Consistent with this, HLA-DR2/DQw1-positive SLE patients have been demonstrated to show low TNF-α production and an increased incidence of lupus nephritis [72]. These results suggest that low TNF-α production may be involved in a genetic predisposition to lupus nephritis. However, another mechanism has also been proposed. Via et al. demonstrated that TNF-α inhibitors promote humoral autoimmunity by selectively inhibiting the induction of cytotoxic T cells that would normally suppress autoreactive B cells [73].

### 3.2. IL-12/23 p40 Inhibitors

Ustekinumab, a fully human mAb against IL-12 and IL-23, which blocks the shared p40 subunit of IL-12 and IL-23, is currently the only approved IL-12/23 p40 inhibitor (Table 1). Ustekinumab is also associated with psoriasiform eruptions and eczematous eruptions, and other adverse reactions, such as lupus-like reactions, sarcoidosis-like reactions, and vitiligo, have also been reported [1].

#### 3.2.1. Psoriasiform Reactions

There have been few case reports of psoriasiform eruptions associated with ustekinumab. Even though this biologic has been in use for a long time, only about 10 cases of psoriasiform eruptions have been reported [1,74]. This may have been due to the fact that the blockade of both Th17 (IL-23) and Th1 (IL-12) responses is implicated in psoriasis caused by ustekinumab. Although the precise mechanism responsible for paradoxical psoriasis during ustekinumab treatment remains unknown, the blockade of both IL-23 and IL-12 pathways by ustekinumab may lead to the overexpression of IFN-α.

#### 3.2.2. Eczematous Reactions

Ustekinumab is also associated with the development of eczematous reactions, although the underlying mechanism is unclear. The Th1/Th17 blockade by ustekinumab may induce a shift toward Th2 response. Following the first case report of such a reaction in a patient with plaque psoriasis and palmoplantar pustular psoriasis [75], a systematic review documented 10 additional cases of ustekinumab-induced eczematous eruptions [76].

#### 3.2.3. Ustekinumab for Skin Reactions Associated with TNF-α Inhibitors

As a human mAb against the p40 subunit shared by the cytokines IL-12 and IL-23, ustekinumab blocks both Th1 and Th17 signaling pathways, which are upregulated in psoriasis [39]. As such, ustekinumab has been shown to be an effective treatment for TNF-a inhibitor-induced paradoxical psoriasis [77,78]. Ezzedine et al. found that the use of ustekinumab completely resolved psoriasiform lesions induced by TNF-α inhibitors in 12 of 14 patients (85.7%) and produced a partial response in the other two cases (14.3%) [77]. In addition, in another study it was reported that TNF-α inhibitor-induced eczematous eruptions were completely resolved by ustekinumab treatment in all 9 cases [77].

### 3.3. IL-17 Inhibitors

Th17 cells are a key T-cell population and act as a proximal regulator of psoriatic skin inflammation. IL-17 inhibitors, including secukinumab, ixekizumab, and brodalumab, are approved for the treatment of moderate-to-severe psoriasis and psoriatic arthritis (Table 1). Secukinumab and ixekizumab are human mAbs targeting IL-17A, whereas brodalumab antagonizes the IL-17RA and disrupts IL-17A, IL-17C, IL-17F, and IL-17A/F heterodimer signaling, giving it potential advantages over selective IL-17A inhibition with secukinumab or ixekizumab [39]. IL-17 inhibitors have been introduced more recently, and the possible cutaneous adverse reactions associated with their use have not been fully elucidated. Numerous cutaneous adverse reactions have been reported to occur in the setting of IL-17 inhibitor treatment, including eczematous eruptions, followed by psoriasiform eruptions, sarcoidosis-like reactions, alopecia areata, and lupus-like reactions [1].

#### 3.3.1. Psoriasiform Reactions

Psoriasiform eruptions have also been reported to be associated with IL-17 inhibitors, particularly secukinumab. A systematic review of clinical reports of new-onset or flare-ups of pre-existing psoriasis induced by biologics revealed 9 cases of paradoxical psoriasis induced by secukinumab [74]. It is hypothesized that the blockade of IL-17A by secukinumab may likewise cause compensatory overproduction of the cytokines produced earlier in this pathway (IL-23 and IL-17F) or of cytokines belonging to the other arm of immunity (IL-12 and TNF-α) [74].

#### 3.3.2. Eczematous Reactions

The most frequently reported cutaneous adverse reactions during IL-17 inhibitor treatment are eczematous eruptions [1]. A case series and a single case of IL-17 inhibitor-induced eczematous eruptions were reported recently [79,80]. Two retrospective studies involving cohorts of 185 and 468 psoriasis patients that were treated with IL-17 inhibitors (either ixekizumab or secukinumab) reported eczematous eruptions prevalence rates of 2.2% and 5.8%, respectively [81,82]. A systematic review of all cases of eczematous eruptions reported in patients treated with biologic agents for psoriasis identified secukinumab as the most frequent cause of such eruptions. It included 92 patients from 24 studies of patients treated with TNF-α inhibitors (adalimumab, etanercept, or infliximab), IL-17 inhibitors (secukinumab or ixekizumab), or IL-12/23p40 inhibitors (ustekinumab) [76]. All 6 biologics were found to have caused eczematous eruptions, with secukinumab being the most common causative drug (46 patients), followed by ixekizumab (15 patients) and ustekinumab (11 patients) [76]. In sharp contrast to secukinumab and ixekizumab, eczematous eruptions caused by brodalumab appear to be very rare. To the best of our knowledge, there has only been one recent report of eczematous eruptions associated with brodalumab [83].

##### Mechanism Responsible for Eczematous Reactions Induced by IL-17 Inhibitors

The mechanism by which IL-17 inhibitors cause eczematous eruptions remains elusive. A history of atopy (atopic eczema, asthma, or allergic rhinoconjunctivitis), which was identified in 11 of 24 cases (46%) in a systematic review, appears to be a risk factor for eczematous eruptions [76,81]. The development of eosinophilia or elevated IgE levels was also documented in 9 of 24 cases (38%) [76].

It is worth noting that there have been a few reports of brodalumab, which targets the IL-17 receptor A, causing eczematous eruptions [83]. In contrast, there have been plenty of reports of eczematous eruptions induced by IL-17A inhibitors, including secukinumab and ixekizumab. This suggests that other IL-17 family members could be responsible for the development of eczematous eruptions, as brodalumab inhibits multiple IL-17 cytokines. The IL-17 cytokine family consists of 6 family members (IL-17A to IL-17F) and receptor subunits (IL-17RA to IL-17RE) [84]. IL-17 receptors are heterodimers composed of IL-17RA associated with either IL-17RC, IL-17RE, or IL-17RB, and the resultant complexes are specific to IL-17A and F, IL-17C, and IL-17E (IL-25), respectively. IL-17E signaling, which is mediated by the IL-17RA/IL-17RB complex, has been reported to be crucial for Th2-mediated inflammation. Similarly, IL-17C signaling, which is mediated by the IL-17RA/IL-17RE complex, has been linked to the pathogenesis of atopic eczema [84]. Therefore, by targeting IL-17RA brodalumab may inhibit the effects of IL-17E and IL-17C, suppressing the development of eczema, unlike ixekizumab or secukinumab, which only target IL-17A [76].

### 3.4. IL-23p19 Inhibitors

IL-23, the “master regulator” of Th17 cells, is a heterodimer composed of two subunits: p40, which is shared with IL-12, and a unique p19 subunit. Recently, IL-23p19 inhibitors, which are specific to IL-23 due to their targeting of the p19 subunit, have started to be used to treat moderate-to-severe psoriasis. These drugs include guselkumab, risankizumab, and tildrakizumab (Table 1). The cutaneous adverse events caused by IL-23p19 inhibitors have not been fully elucidated, and only a few sporadic cases of eczematous reactions have been reported.

#### 3.4.1. Eczematous Reactions

There have been few reports about IL-23p19 inhibitors causing eczematous eruptions, presumably because these drugs are the most recently approved biologics for psoriasis. On the other hand, it was suggested that IL-23 inhibitors may cause eczematous reactions less often than IL-17A inhibitors [1], as an analysis of clinical trial data did not show any reported eczematous reactions [85]. Only four cases of eczematous eruptions that may have been induced by IL-23p19 inhibitors in psoriasis patients have been described in case reports [86,87,88,89] (Table 2). Three cases involved guselkumab [86,87,88], and one involved risankizumab [89]. To the best of our knowledge, tildrakizumab has not been reported to cause eczematous reactions. As is the case for other classes of biologics, an underlying atopic predisposition may play a role in eczematous eruptions caused by IL-23p19 inhibitors, as all of the reported patients except for patient 3 had a history of atopy (Table 2).

##### Mechanism Responsible for Eczematous Reactions Induced by IL-23p19 Inhibitors

The mechanisms responsible for eczematous eruptions that arise secondary to IL-23p19 inhibitor treatment have not been studied in detail, probably due to the recent introduction of these drugs as therapeutics for psoriasis. As DCs produce IL-23, driving Th-cell differentiation toward Th17 cells, and Th17 cells secrete TNF-α as well as IL-17 under the regulation of IL-23, it is hypothesized that IL-23p19 inhibitors act as IL-17 inhibitors and partially as TNF-α inhibitors. IL-17, the major effector cytokine in psoriasis, acts alone or synergistically with TNF-α to induce the expression and release of many psoriasis-related proteins from keratinocytes, such as β-defensin 4 and S100A7 [19]. Thus, eczematous eruptions associated with IL-23p19 inhibitors could also be explained by the concept of a Th1/Th17 to Th2 switch, i.e., targeting the Th1/Th17 axis could provoke a shift towards Th2-driven immune responses, and, thus, lead to flares of atopic eczema [9,76].

### 3.5. IL-4Rα Inhibitors

Currently, dupilumab, an IL-4Rα inhibitor, is the only approved biologic agent for moderate-to-severe AD in adults and children [1,90] (Table 1). Multiple clinical trials have demonstrated the efficacy and safety of dupilumab against moderate-to-severe AD [91,92]. Dupilumab is a fully human mAb directed against the IL-4Rα subunit, which blocks signaling by both IL-4 and IL-13, key mediators of type 2 inflammation. The following dupilumab-induced PRs have been reported in the literature.

#### 3.5.1. Psoriasiform Reactions

Since the first case reported by Tracey et al., several further sporadic case reports have suggested that dupilumab may lead to the development of psoriasis [93,94,95,96]. Subsequently, Napolitano et al. reported that psoriasiform reactions developed during dupilumab treatment in 3 out of 90 adult AD patients (3.33%) [97]. Several large cohort studies have since been performed, demonstrating low incidence rates (a few percent) of new-onset psoriasis among dupilumab-treated patients. For instance, two retrospective studies of 165 and 373 AD patients treated with dupilumab demonstrated that 5 patients (3.03%) [98] and 7 patients (1.88%) [99] developed psoriasiform eruptions, respectively. It is now considered that psoriasiform eruptions are one of the most frequently observed PRs associated with dupilumab [1]. The mean latency period after the initiation of dupilumab therapy was 16 weeks [99]. Similarly, a systematic review of published cases, which included 47 patients from 26 studies, revealed that the mean latency period from the initiation of dupilumab to psoriasis onset was 3.7 months [100]. Most reported cases resolved after the discontinuation of dupilumab and/or the initiation of topical corticosteroids.

##### Mechanism Responsible for Psoriasiform Reactions Induced by IL-4Rα Inhibitors

The mechanisms responsible for the development of psoriasis during dupilumab therapy in AD patients are not well known. One possible explanation is that a genetic predisposition could exist in the AD patients that develop psoriasis [99]. Another possibility, which is the most well-described mechanism, is the induction of immune deviation, i.e., antagonism of the Th2 pathway by dupilumab may lead to an opposing shift toward the Th1 and Th17 pathway, causing Th1/Th17-mediated disorders, such as psoriasis [4,9]. A prospective dose-escalation study of human IL-4, which included 20 patients with severe psoriasis, revealed that IL-4 treatment markedly improved psoriasis and induced a pronounced skewing toward Th2 cytokine production in both psoriatic skin lesions and the circulation [61]. These results fit with the concept of mutual T-cell antagonism causing psoriasis and AD [9]. Recent studies have suggested another mechanism, in which IL-4 acts as a regulatory cytokine and abrogates Th17 responses by selectively preventing IL-23 transcription and secretion in antigen-presenting cells, while sparing IL-12-dependent Th1 responses [101]. Indeed, quantitative real-time polymerase chain reaction analysis and immunofluorescence staining of the lesional skin of AD patients who developed psoriasiform reactions during dupilumab treatment demonstrated increased expression of IL-23A, suggesting the activation of the Th17 pathway [102]. In another study, skin biopsy samples from an AD patient who developed psoriasiform eruptions 8 months after dupilumab treatment were subjected to RNA in situ hybridization (RISH) [103]. RISH performed before the dupilumab treatment showed high IL-13 expression, but IL-17A was undetectable. However, an analysis of the dupilumab-induced psoriasiform eruptions revealed de novo IL-17A expression, whereas IL-13 expression was diminished [103].

#### 3.5.2. Eczematous Reactions

Some patients who are treated with dupilumab develop eczematous PRs involving regional or localized dermatitis, most commonly on the face, the periocular region, and/or neck [1,90]. A retrospective cohort study by Zhu et al. reported that of 73 AD patients who received dupilumab therapy, 17 (23%) developed new regional dermatoses, with facial involvement seen in 14 cases [104]. The proposed causative mechanisms of such dupilumab-induced head and neck dermatitis include unrecognized allergic contact dermatitis, for which patch testing may be diagnostically useful, and unopposed activation of the Th1/Th17 pathway [90,104,105], which may cause rosacea flares [106]. An AD flare due to topical steroid withdrawal during dupilumab therapy may also explain the occurrence of such head and neck dermatitis [105].

## 4. Conclusions

Deep insights into the molecular mechanisms involved in inflammatory immune diseases have accelerated the development of targeted therapies. As biologic agents will be used far more in the future, it is essential to better understand the mechanisms underlying any adverse reactions they can produce. Our analysis of the adverse reactions to different biologic agents revealed that many appear to be related to the biological activity of the agents and are not due to immune responses against them, as occurs in hypersensitivity. Further research will be needed to elucidate the detailed molecular and genetic bases of such reactions and identify any predisposing factors. Such information would help to guide future treatment strategies for preventing these adverse reactions by stratifying individuals according to risk and ultimately move us toward the goal of personalized medicine.

## Figures and Tables

**Figure 1 biomedicines-10-01485-f001:**
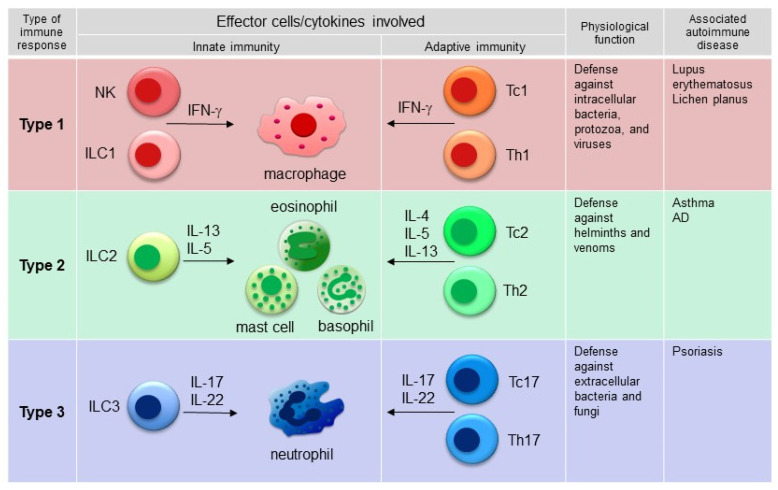
Three major types of immune response pattern and related autoimmune diseases. Type 1 immunity involves IFN-γ-producing Th1 cells, group 1 innate lymphoid cells (ILC1s), Tc1 cells, natural killer (NK) cells, and macrophages and provides protection against intracellular microbes, such as bacteria, protozoa, and some viruses. Type 2 immunity involves Th2 cells, ILC2s, Tc2 cells, eosinophils, basophils, and mast cells, which produce type 2 cytokines, such as IL-4, IL-5, and IL-13, and mainly provides protection against helminths and venoms. Type 3 immunity is characterized by Th17 cells, ILC3s, and Tc17 cells, which produce IL-17 and IL-22, and provide protection against extracellular bacteria and fungi. Examples of autoimmune diseases associated with each type of immune response are depicted.

**Figure 2 biomedicines-10-01485-f002:**
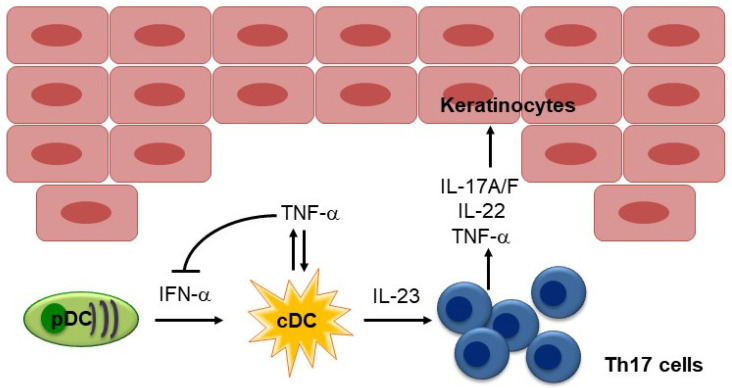
Pathogenesis of psoriasis. The characteristic TNF-IL-23-Th17 pathway in psoriasis is illustrated. In the early phase of psoriatic skin lesion, pDC-IFN pathway plays a role. TNF-α inhibits IFN-α production by pDCs. pDC, plasmacytoid DC; cDC, conventional DC.

**Table 1 biomedicines-10-01485-t001:** Biologic agents used to treat psoriasis and AD.

Indications	Class	Medication	Description
Psoriasis	TNF-α inhibitors	Infliximab	chimeric mAb to TNF-α
		Adalimumab	fully human mAb to TNF-α
		Certolizumab pegol	humanized mAb to TNF-α
		Etanercept	recombinant human TNF-R/IgGFc
	IL-17 inhibitors	Secukinumab	fully human mAb to IL-17A
		Brodalumab	fully human mAb to IL-17RA
		Ixekizumab	humanized mAb to IL-17A
	IL-12/23p40 inhibitor	Ustekinumab	fully human mAb to IL-12/23p40
	IL-23p19 inhibitors	Guselkumab	fully human mAb to IL-23p19
		Risankizumab	humanized mAb to IL-23p19
		Tildrakizumab	humanized mAb to IL-23p19
AD	IL-4Rα inhibitor	Dupilumab	fully human mAb to IL-4Rα

mAb, monoclonal antibody; R, receptor.

**Table 2 biomedicines-10-01485-t002:** Clinical details of cases of eczematous eruptions secondary to IL-23p19 inhibitor treatment.

Pt No.	Author	Age	Sex	Clinical Description (Duration)	Biologic	Time of Onset	Previous Atopy	Histology	Clinical Course
1	Reyn et al. (2019)	47	M	Psoriasis vulgaris (NR)	Guselkumab	10 w	AD	Acanthosis, spongiosis, lymphocytic inflammatory infiltrates mixed with eosinophils	Guselkumab discontinued; Eczema resolved with tar preparation
2	Truong et al. (2019)	40	M	Pustular psoriasis (since childhood)	Guselkumab	3 m	NR	Psoriasiform epidermal hyperplasia, parakeratosis, spongiosis, perivascular lymphohistiocytic infiltrates	NR
3	Miyagawa et al. (2021)	75	M	Pustular psoriasis (13 years)	Guselkumab	3 m	None	Parakeratosis, spongiosis, perivascular inflammatory infiltrates consisting of lymphocytes and eosinophils	Switched from secukinumab due to eczematous eruptions; Guselkumab continued; Treated with topical corticosteroids; Eczema persisted
4	Kromer et al. (2020)	52	M	Psoriasis vulgaris (4 years)	Risankizumab	3 w	Allergic rhinitis	Acanthosis, compact orthokeratosis with foci of parakeratosis, spongiosis, perivascular lymphocytic infiltrates	Guselkumab continued; Improved after treatment with topical corticosteroids
5	as above	59	M	Psoriasis vulgaris (26 years)	Risankizumab	4 w	Allergic rhinitis, Asthma	NR	Switched to ustekinumab; Eczema improved

Pt, patient; M, male; w, weeks; m, months; NR, not reported.

## Data Availability

Not applicable.

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
