# Peer review of "Pathogenesis of Paradoxical Reactions Associated with Targeted Biologic Agents for Inflammatory Skin Diseases"

_biomedicines, 2022, doi:10.3390/biomedicines10071485_

Round 1

Reviewer 1 Report

Is the histologic pattern of PRs similar or equal to the principal diseases? Do the eosinophils appear in the great amount? How long do the side effects last after the witdrawl of pathogenetic agents? Do you know some predisposing factors?

Reviewer 2 Report

The review paper is comprehensive and concise. All of the considerable points in the topic are included. The authors managed to gather the data existing on the paradoxical reactions caused by biologic therapy for inflammatory skin diseases and to establish valid conclusions.

No major flaws are associated with this paper.

Minor points:

1. If possible, add illustrative material for the second part of the review
